# Characterization of Flowering Time and Pollen Production in Jojoba (*Simmondsia chinensis*) towards a Strategy for the Selection of Elite Male Genotypes

**Noemi Tel-Zur** [1,*] , **Ronen Rothschild** [2], **Udi Zurgil** [1] **and Yiftach Vaknin** [3]

[1] French Associates Institute for Agriculture and Biotechnology of Drylands, The Jacob Blaustein Institutes for Desert Research, Ben-Gurion University of the Negev, Sede Boqer Campus, Beersheba 84990000, Israel; udizur@gmail.com

[2] Jojoba Israel Ltd., Kibbutz Hatzerim 8542000, Israel; ronen.rotschild@hatzerim.co.il

[3] Institute of Plant Sciences, Agricultural Research Organization (ARO), Volcani Center, Rishon LeZion 7505101, Israel; yiftachv@volcani.agri.gov.il

\* Correspondence: telzur@bgu.ac.il

**Abstract:** The seeds of the dioecious shrub jojoba (*Simmondsia chinensis* (Link) Schneider) yield a liquid wax that is in high demand for the cosmetics industry. While elite female cultivars of this species are currently clonally propagated, male plants are grown from seed, resulting in large variations in both the flowering period and the pollen viability, and hence large variation in yields. We characterized the existing male plant material in a local plantation as a platform for future selection of elite male cultivars that would produce sufficient amounts of viable pollen throughout the extended flowering period of the female cultivars. Using as a guide the number of viable pollen grains per 1-m branch, defined here as the calculated effective pollen productivity (EPP), we identified plants with an elevated EPP that flower concurrently with the female cultivars.

**Keywords:** dioecious; flowering time; phenological diversity; pollen viability

---

## 1. Introduction

Jojoba (*Simmondsia chinensis* (Link) Schneider) is a dioecious, wind-pollinated, perennial shrub that is cultivated for the liquid wax produced in its seeds [1,2]. The high demand for the wax in the cosmetics industry is one of the factors that has been driving efforts to improve yields and hence profitability [3]. Following a selection program for female cultivars producing large amounts of seeds [4,5], there has been a marked expansion of planted areas, particularly in drylands [6]. However, any further increases in yields will require a steady supply of high-quality pollen throughout the extended female flowering season [7].

While elite female cultivars are currently clonally propagated from stem cuttings, male plants are grown from seed, which results in large variations in flowering periods and in pollen viability [8] (the process involves germinating the seeds, growing the plants in pots until the first flowering season, and only then transferring the male plants to the orchard). In Israel, female cultivars usually bloom from the middle of February to late March, but some bloom early, and some late. In the past few years, a delay in blooming time has been observed, probably due to the late arrival of spring or to global warming. This variability in the female blooming period and the potential for a poor overlap with male blooming stress the importance of identifying males with early, middle, and late blooming characteristics. Thus, with the aim of establishing a platform for further breeding and selection of elite male cultivars that could be vegetatively propagated, we set out to characterize the reproductive

performances, i.e., flowering phenology, pollen production, and pollen viability, of the male genotypes currently under cultivation in the Hatzerim plantation.

## 2. Materials and Methods

The study was conducted from 2016 to 2018, i.e., over three years, on 8- and 25-year-old male shrubs (n = 150 and 150, respectively) growing in the Hatzerim plantation in Israel's Negev Desert (31.2406° N 34.7174° E). The annual irrigation rate was 7200–7500 m$^3$ of recycled water, which was supplemented with fertilization to provide 250, 100, and 350 kg of N (nitrogen), P (phosphorus), and K potassium), respectively, per hectare.

Pollen production levels were recorded in 2016 and 2017, and the flowering phenology was recorded in three consecutive years (2016–2018). Peak flowering for the male genotypes was arbitrarily designated as "early" (before), "middle" (concurrent with), or "late" (after) relative to the peak flowering of the female plants. The plants in the Hatzerim plantation are pruned at the end of each summer, and the male inflorescences then develop along newly emerging branches. Thus, inflorescence density and production of viable pollen (i.e., number of viable pollen grains produced per 1 m of branch) were recorded for new branches. Six branches were sampled for each shrub (three branches from its eastern face and three from its western face, where the shrubs are planted in rows aligned north to south). The length and number of inflorescences for each of these branches were recorded, as were the numbers of flowers per inflorescence in 10 fully developed inflorescences (n = 10). To determine the number of pollen grains per flower, 6 to 12 flowers per studied male genotype were harvested before stamen dehiscence, and at anthesis, the anthers of each flower were transferred to a micro-tube containing 300 μL of 0.5 M sucrose (n = 6–12). The pollen suspension was injected into a hemocytometer, and the total number of pollen grains per flower was determined three times per flower for three flowers. Pollen stainability/viability was assayed using 2% acetocarmine. At least 300 pollen grains were examined per flower in three flowers per genotype. The number of viable pollen grains per 1 m branch, designated here as "effective pollen productivity" (EPP), was then calculated as the number of inflorescences per 1 m branch × the number of flowers per inflorescence × the number of viable pollen grains per flower. The data obtained on flowering phenology and pollen production during 2016 facilitated the selection of the 11 best-performing genotypes for further evaluation during 2017. The flowering period of these 11 male genotypes was followed in both 2017 and 2018.

Statistical analyses were performed with JMP 14.0.0 software (SAS Institute Inc. Cary, NC, USA): one-way analysis of variance (ANOVA) and post-hoc Tukey–Kramer HSD (honestly significant difference) test were applied to compare reproductive traits among the 11 selected lines in 2016 and 2017. Pearson correlation analyses were used to evaluate separately the relationships between the same reproductive traits in 2016 and 2017 among the 11 selected lines and between different reproductive traits in 2016 and 2017. All data are presented as means ± standard error. Values are reported as significantly different if $p < 0.05$.

## 3. Results and Discussion

Successful cultivation of dioecious species depends on good synchronization of the flowering periods of the female and male plants in a plantation ([9] and references therein). In jojoba, variations in yield have been attributed to pollination constraints, and it has been suggested that pollen supplementation could increase yields.

Our results revealed extensive variability among the male genotypes (probably reflecting the variability in their genetic backgrounds), which we were able to exploit for the identification of male shrubs with above-average reproductive traits and reliable flowering periods (Table 1). Among the 21 male genotypes studied in 2016 (Figure 1), nine bloomed early (first two weeks in February); nine were middle bloomers (mid-February to mid-March); and three bloomed late (after mid-March) (Table 1). A follow-up of the flowering periods in 2017 and 2018 for the 11 subsequently chosen genotypes revealed a constant and reliable flowering period in only six (Table 1).

**Table 1.** Flowering and floral traits of selected male jojoba seedlings in 2016, 2017, and 2018.

| Line | Inflorescences/m | | Flowers/Inflorescence | | Pollen/Flower | | Viability (%) | | Viable Pollen/m (EPP) | | Flowering | | |
|---|---|---|---|---|---|---|---|---|---|---|---|---|---|
| | 2016 | 2017 | 2016 | 2017 | 2016 | 2017 | 2016 | 2017 | 2016 | 2017 | 2016 | 2017 | 2018 |
| 40–39^ | 74.3 ± 13.6 | 38.7 ± 3.0 | 21.2 ± 0.6 | 20.8 ± 1.4 | $5.5 \times 10^5 \pm 3.8 \times 10^4$ | $4.2 \times 10^5 \pm 3.2 \times 10^4$ | 93.4 ± 1.6 | 91.2 ± 1.7 | $8.0 \times 10^8$ | $3.1 \times 10^8$ | E | E | M |
| 40–11^ | 79.2 ± 7.0 | 60.0 ± 4.4 | 18.2 ± 1.2 | 21.2 ± 1.4 | $5.2 \times 10^5 \pm 1.4 \times 10^4$ | $4.6 \times 10^5 \pm 4.3 \times 10^4$ | 81.4 ± 4.7 | 91.7 ± 1.2 | $6.1 \times 10^8$ | $5.3 \times 10^8$ | E | M | M |
| 35–11*^ | 58.7 ± 4.5 | 42.6 ± 7.6 | 15.8 ± 0.6 | 14.7 ± 0.7 | $5.9 \times 10^5 \pm 4.0 \times 10^4$ | $4.4 \times 10^5 \pm 6.5 \times 10^4$ | 50.8 ± 10.9 | 81.0 ± 3.9 | $2.8 \times 10^8$ | $2.2 \times 10^8$ | E | E | E |
| 40–60*^ | 46.8 ± 4.3 | 31.8 ± 5.5 | 22.6 ± 2.3 | 14.3 ± 1.2 | $4.5 \times 10^5 \pm 3.3 \times 10^4$ | $3.8 \times 10^5 \pm 1.9 \times 10^4$ | 66.3 ± 7.7 | 92.7 ± 2.7 | $3.1 \times 10^8$ | $1.6 \times 10^8$ | E | E | E |
| 20–20+ | 44.2 ± 12.0 | | 15.0 ± 0.8 | | $5.9 \times 10^5 \pm 6.3 \times 10^4$ | | 67.5 ± 1.0 | | $2.6 \times 10^8$ | | E | | |
| 40–20^ | 37.2 ± 4.9 | | 24.2 ± 2.4 | | $4.0 \times 10^5 \pm 2.4 \times 10^4$ | | 81.7 ± 1.0 | | $3.0 \times 10^8$ | | E | | |
| 40–58^ | 24.3 ± 2.0 | | 17.7 ± 0.9 | | $6.7 \times 10^5 \pm 3.8 \times 10^4$ | | 91.1 ± 2.6 | | $2.6 \times 10^8$ | | E | | |
| 35–57^ | 37.1 ± 3.8 | | 14.6 ± 0.5 | | $4.1 \times 10^5 \pm 1.7 \times 10^4$ | | 91.2 ± 1.1 | | $2.0 \times 10^8$ | | E | | |
| 20–14+ | 39.9 ± 9.6 | | 13.1 ± 1.1 | | $3.7 \times 10^5 \pm 3.0 \times 10^4$ | | 56.9 ± 19.8 | | $1.0 \times 10^8$ | | E | | |
| 40–51^ | 103.3 ± 10.3 | 56.0 ± 5.9 | 14.9 ± 1.0 | 17.3 ± 1.6 | $5.5 \times 10^5 \pm 5.5 \times 10^4$ | $4.9 \times 10^5 \pm 2.2 \times 10^4$ | 84.5 ± 2.9 | 85.8 ± 3.9 | $7.1 \times 10^8$ | $4.1 \times 10^8$ | M | M | M |
| 40–14*^ | 71.1 ± 12.6 | 34.3 ± 4.7 | 24.7 ± 2.0 | 27.5 ± 2.6 | $4.8 \times 10^5 \pm 2.5 \times 10^4$ | $4.2 \times 10^5 \pm 2.5 \times 10^4$ | 87.9 ± 6.4 | 92.6 ± 1.6 | $7.4 \times 10^8$ | $3.6 \times 10^8$ | M | M | M |
| 35–03*^ | 68.0 ± 7.1 | 45.0 ± 7.0 | 23.1 ± 1.4 | 18.1 ± 1.3 | $5.2 \times 10^5 \pm 3.4 \times 10^4$ | $4.0 \times 10^5 \pm 3.7 \times 10^4$ | 86.1 ± 3.2 | 84.9 ± 1.9 | $7.0 \times 10^8$ | $2.8 \times 10^8$ | M | M | M |
| 40–30^ | 74.0 ± 13.0 | 42.2 ± 10.2 | 20.7 ± 0.8 | 18.3 ± 1.2 | $4.5 \times 10^5 \pm 3.2 \times 10^4$ | $3.8 \times 10^5 \pm 1.7 \times 10^4$ | 89.2 ± 5.2 | 90.5 ± 2.6 | $6.2 \times 10^8$ | $2.6 \times 10^8$ | M | M | L |
| 35–36^ | 52.7 ± 5.7 | | 20.4 ± 1.2 | | $6.1 \times 10^5 \pm 6.4 \times 10^4$ | | 68.4 ± 5.4 | | $4.5 \times 10^8$ | | M | | |
| 40–49^ | 54.5 ± 6.8 | | 30.5 ± 1.1 | | $3.6 \times 10^5 \pm 3.0 \times 10^4$ | | 76.6 ± 8.0 | | $4.5 \times 10^8$ | | M | | |
| 20–31+ | 52.3 ± 11.1 | | 14.0 ± 1.7 | | $5.9 \times 10^5 \pm 4.7 \times 10^4$ | | 91.4 ± 2.0 | | $4.0 \times 10^8$ | | M | | |
| 35–76^ | 63.2 ± 11.1 | | 17.5 ± 0.8 | | $3.7 \times 10^5 \pm 1.7 \times 10^4$ | | 87.9 ± 3.9 | | $3.6 \times 10^8$ | | M | | |
| 35–25^ | 42.5 ± 4.9 | | 19.2 ± 1.4 | | $3.9 \times 10^5 \pm 1.9 \times 10^4$ | | 72.3 ± 6.9 | | $2.3 \times 10^8$ | | M | | |
| 35–18*^ | 58.4 ± 8.1 | 34.5 ± 6.0 | 24.7 ± 1.6 | 20.9 ± 1.1 | $7.0 \times 10^5 \pm 4.3 \times 10^4$ | $4.3 \times 10^5 \pm 1.3 \times 10^4$ | 71.5 ± 8.4 | 82.5 ± 3.4 | $7.2 \times 10^8$ | $2.6 \times 10^8$ | L | L | L |
| 35–51^ | 62.6 ± 8.9 | 25.2 ± 2.7 | 14.7 ± 1.6 | 18.1 ± 1.1 | $6.0 \times 10^5 \pm 4.4 \times 10^4$ | $5.2 \times 10^5 \pm 5.0 \times 10^4$ | 88.4 ± 3.2 | 89.9 ± 1.4 | $4.9 \times 10^8$ | $2.1 \times 10^8$ | L | L | E |
| 40–65^ | 57.3 ± 9.3 | 36.2 ± 6.6 | 9.1 ± 0.7 | 11.9 ± 0.8 | $3.8 \times 10^5 \pm 1.8 \times 10^4$ | $3.7 \times 10^5 \pm 4.0 \times 10^4$ | 62.5 ± 3.6 | 92.6 ± 4.4 | $1.3 \times 10^8$ | $1.5 \times 10^8$ | L | L | M |

E: early, M: middle, L: late. *selected lines for future vegetative propagation. ^8 and +25 years old at the beginning of the study.

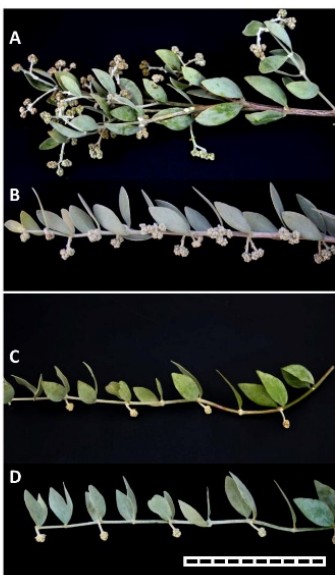

**Figure 1.** Inflorescence density in (**A**,**B**) selected male genotypes and (**C**,**D**) regular lines with low performances. Bar: 10 cm.

The 21 male genotypes studied in 2016 showed a wide variability in the number of inflorescences per 1 m branch ($24.3 \pm 2.0$ to $103.3 \pm 10.3$), the number of flowers per inflorescence ($9.1 \pm 0.7$ to $30.5 \pm 1.1$), the number of pollen grains per flower ($3.6 \times 10^5 \pm 3.0 \times 10^4$ to $7.0 \times 10^5 \pm 4.3 \times 10^4$), and the pollen viability ($50.8\% \pm 10.9\%$ to $93.4\% \pm 1.6\%$). The EPP was therefore also highly variable ($1.0 \times 10^8$ to $8.0 \times 10^8$). Among the 11 genotypes followed up in 2017, the variability was likewise high for the number of inflorescences per 1 m branch ($25.2 \pm 2.7$ to $60.0 \pm 4.4$), the number of flowers per inflorescence ($11.9 \pm 0.8$ to $27.5 \pm 2.6$), and the number of pollen grains per flower ($3.7 \times 10^5 \pm 4.0 \times 10^4$ to $5.2 \times 10^5 \pm 5.0 \times 10^4$). An exception was pollen viability, which was relatively high and less variable ($81.0\% \pm 3.9\%$ to $92.7\% \pm 2.7\%$). Nevertheless, the calculated EPP remained highly variable ($1.5 \times 10^8$ to $5.3 \times 10^8$), probably as a result of the above-described variation in pollen production.

A comparison between the data sets for 2016 and 2017 for the 11 studied genotypes revealed that in 2017, the plants produced fewer inflorescences with fewer pollen grains per flower; however, the number of flowers per inflorescence remained similar, while pollen viability was slightly increased (Table 2). The calculated EPP was highly variable both in 2016 and 2017, with values ranging from $1.0 \times 10^8$ to $8.0 \times 10^8$ in 2016 and from $1.5 \times 10^8$ to $5.3 \times 10^8$ in 2017 (Table 1). The calculated EPP was higher in 2016 than in 2017, probably due to the significantly higher production of inflorescences and pollen grains per flower (Table 2). These differences may be attributed to the plants' responses to the environmental conditions of the particular year and may also reflect natural variations in alternating years.

**Table 2.** Comparison between 2016 and 2017 of male traits in 11 selected genotypes.

| Male trait | 2016 (means ± SE, $n = 11$) | 2017 (means ± SE, $n = 11$) | *P* (*t*-test) |
|---|---|---|---|
| Inflorescences/m | $68.52 \pm 4.50$ A | $40.58 \pm 3.09$ B | <0.0001 |
| Flowers/inflorescence | $19.06 \pm 1.50$ A | $18.46 \pm 1.27$ A | 0.7642 |
| Pollen/flower | $5.3 \times 10^5 \pm 2.6 \times 10^4$ A | $4.3 \times 10^5 \pm 1.4 \times 10^4$ B | 0.0036 |
| Pollen viability (%) | $78.4 \pm 4.1$ B | $88.7 \pm 1.3$ A | 0.0263 |
| Viable pollen/m (EPP) | $5.5 \times 10^8 \pm 6.7 \times 10^7$ A | $2.9 \times 10^8 \pm 3.4 \times 10^7$ B | 0.0020 |

Different letters indicate significant differences. SE: standard error.

In 2016 and 2017, EPP was positively correlated with the number of inflorescences per 1 m branch, the number of flowers per inflorescence, and the number of pollen grains per flower (Table 3). This positive correlation indicated a strategy by which the number of pollen grains per plant could be enhanced by increasing all stages of the reproductive growth in the plant. An examination of the other correlations for pollen viability revealed a positive correlation with the number of inflorescences per 1 m branch in 2016 but a lack of correlation with EPP in 2017 (Table 3). This lack of correlation in 2017 suggests that the contribution of pollen viability to EPP may change dramatically between years and therefore should be further investigated. Correlation analysis for male traits revealed highly positive correlations for the number of inflorescences per 1 m branch ($R = 0.72$), the number of flowers per inflorescence ($R = 0.65$), the number of pollen grains per flower ($R = 0.65$), and the EPP ($R = 0.62$), but not for pollen viability ($R = 0.13$), which may have been affected by the environmental conditions in the years of the study. The finding that the plants "invested" more in pollen quality than in quantity in a year of reduced pollen production, such as 2017, may provide a clue to a strategy that allows the plants to compensate, to some extent, for the reduced level of pollen production. Just as environmental and growth conditions affect pollen traits, so are nutrient levels linked to pollen quality and productivity, as has recently been reported in *Juniperus communis* by Pers-Kamczyc et. al. [10].

**Table 3.** Correlation matrixes highlighting significant correlations (*R*-values) between properties within genotypes in 2016 and 2017 (*n* = 11).

| 2016 | Inflorescences/m | Flowers/Inflorescence | Pollen/Flower | Viability (%) |
|---|---|---|---|---|
| Flowers/inflorescence | - | | | |
| Pollen/flower | - | - | | |
| Viability (%) | 0.56 | - | - | |
| Viable pollen/m (EPP) | 0.60 | 0.64 | 0.42 | 0.79 |
| **2017** | | | | |
| Flowers/inflorescence | - | | | |
| Pollen/flower | - | - | | |
| Viability (%) | - | - | - | |
| Viable pollen/m (EPP) | 0.77 | 0.61 | 0.40 | - |

(-): non-significant correlation. EPP—effective pollen productivity.

## 4. Conclusions and Future Breeding Strategies

Our study covering three flowering seasons enabled us to identify several male genotypes with high EPP values that reliably maintained their flowering phenology as early, middle or late bloomers (Table 1). Five of them have already been selected for vegetative propagation. A long-term study will now be started to verify our premise that these traits will persist when these plants are clonally propagated. Plantation design with such elite male lines that will ensure that pollen availability throughout the female flowering season will improve pollination rates and hence increase yields.

Our study revealed that the best strategy to identify males as potential elite cultivars is not to focus on a single reproductive trait, such as pollen viability or number of inflorescences, but to acquire data on all aspects of their reproductive traits as a means to providing a more accurate measure of their reproductive potential. We found that availability of viable pollen in a particular year was directly dependent on all the reproductive traits in that year, and on most, but not all, in the subsequent year (Table 3). A study of multiple reproductive traits—versus only a single trait, such as pollen viability—will enable the breeder to consistently and reliably identify elite males as future pollen providers.

Now that we have shown the potential of our breeding methodology for identifying prospective elite male cultivars, growers are in a position to expand future breeding programs over a much broader range of male genotypes, under several climatic regions, both in Israel and in other countries. Concurrently, as our group identifies prospective elite males with elevated EPP and desirable bloom

phenologies, we intend to embark on a long-term research project that would test the males against at least three of our best-performing female cultivars. We will focus not only on EPP and pollen availability but also on pollen germination on the stigma, fruit and seed sets, and wax chemical traits (i.e., xenia and metaxenia effects). It is our firm belief the such a program would eventually lead to the realization of the potential for enhanced yield quantity and quality.

**Author Contributions:** N.T.-Z. and Y.V. conceived the project. N.T.-Z., U.Z., R.R., and Y.V. collected data and conducted the analyses. N.T.-Z. and Y.V. wrote the manuscript. All authors read and agreed to the published version of the manuscript.

**Funding:** This research was supported by the Chief Scientist, Ministry of Agriculture and Rural Development, project number 20-13-0025.

**Conflicts of Interest:** The authors declare no conflict of interest.

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
