# Peer review of "Characterization of Flowering Time and Pollen Production in Jojoba (Simmondsia chinensis) towards a Strategy for the Selection of Elite Male Genotypes"

_agronomy, doi:10.3390/agronomy10040592_

Round 1
Reviewer 1 Report
Revised manuscript, can now be approved.
Reviewer 2 Report
Authors addressed all my comments and concerns properly. Also, they have improved the manuscript.
This manuscript is a resubmission of an earlier submission. The following is a list of the peer review reports and author responses from that submission.
Round 1
Reviewer 1 Report
The authors have studied traits related to pollen production in dioecious Jojoba (Simmondsia chinensis), with the aim of having more stable male lines in the future, in order to ensure the production of quality seed for cosmetic industry purposes. The manuscript is mainly well written and clear.
I have the following comments:
lines 52-53: what does the N=10 refer to?
First selection was based on 1 year performance. Considering the variation detected between years, would the authors still do the selection after only one year?
Would be good to have EPP values in the table as well (table 1). A more detailed description of the selection criteria please. Purely based on EPP ranking, with exclusion of very early and very late flowering lines?
Did the EPP ranking of lines change between years?
Lines 108-110: “In 2016 and 2017, EPP was positively correlated with the number of inflorescences per 1-m 108 branch, the number of flowers per inflorescence, and the number of pollen grains per flower (Table 109 3).” The formula of EPP includes those traits (total amount of viable pollen per shrub, right?), so of course there is correlation?! I don’t understand the meaning of these correlation tests.
Lines 115-118: “The finding that 115 the plants 'invested' more in pollen quality than in quantity in a year of reduced pollen production, 116 such as 2017, may provide a clue to a strategy that allows the plants to compensate, to some extent, 117 for the reduced level of pollen production.”. This is an interesting topic which may apply to other biochemical processes as well. There is some indication that the amount of pollen allergens in “fixed at tree level”, i.e., the more pollen produced, the smaller the allergen content of pollen grains.
Table 3: please add information on the number of correlation tests (N).
I would have liked to read the authors’ thoughts about future research aspects. Especially discussion on (i) how male*female gametophytic interaction /compatibilities (/female choice) might make the breeding programmes more complex; and (ii) how to assure a wide enough genetic variation for future breeding needs.
Author Response
Reviewer #1
The authors have studied traits related to pollen production in dioecious Jojoba (Simmondsia chinensis), with the aim of having more stable male lines in the future, in order to ensure the production of quality seed for cosmetic industry purposes. The manuscript is mainly well written and clear.
I have the following comments:
Lines 52-53: what does the N=10 refer to?
- Ten biological replicates. Number of flowers in ten inflorescences.
First selection was based on 1 year performance. Considering the variation detected between years, would the authors still do the selection after only one year?
- We did not undertake a selection program. Rather, we report in this manuscript a first screen/characterization of male genotypes as a platform for further selection. During the first year we chose and focused our research on 21 male genotypes among the 300 available for this study. The choice was based on visual differences in the number and density of male inflorescences (see Figure 1).
Would be good to have EPP values in the table as well (table 1). A more detailed description of the selection criteria please. Purely based on EPP ranking, with exclusion of very early and very late flowering lines?
Did the EPP ranking of lines change between years?
- The EPP ranking of the lines is given in Table 1 under "Viable pollen/m". To clarify this point, we added "(EPP)" in parentheses: "Viable pollen/m (EPP)". The EPP ranking of the lines changed between the years and this finding is addressed in lines 112-113 and is also shown in Table 2.
Lines 108-110: “In 2016 and 2017, EPP was positively correlated with the number of inflorescences per 1-m branch, the number of flowers per inflorescence, and the number of pollen grains per flower (Table 3).” The formula of EPP includes those traits (total amount of viable pollen per shrub, right?), so of course there is correlation?! I don’t understand the meaning of these correlation tests.
- We agree that sometimes correlations show connections that at first glance seem obvious, but upon closer inspection they reveal important relationships and changes between the years, which shed new light on our data. Here, we wanted to stress that the elevated EPP in some lines was not simply the result of an elevated number of inflorescences, an elevated number of flowers per inflorescence, or an elevated number of pollen grains per flower, or even enhanced pollen viability, but the combined result of all traits together. Any one of the measured traits could have been the factor causing the elevated EPP and therefore elevated pollen availability, which is our desired trait. A strategy of elevating all reproductive traits at the same time suggests a broad investment in all stages of reproductive development to enhance its potential. Also, the absence of significant correlations, such as between number of inflorescences and pollen viability in 2017, but not in 2016, reveals that effect of pollen on EPP in some years may not be significant. This finding must therefore impact the reproductive strategy for the plant. Therefore, we added: "This lack of correlation in 2017 suggests that the contribution of pollen viability to EPP may change dramatically between years, and therefore should be further investigated" in lines 125-127.
Additionally, we found differences between the years in the level of correlation and particularly in the absence of correlation of EPP with pollen viability in 2017 versus 2016, which shows that some traits are more stable than others in determining the level of EPP. To further clarify that we added: "This positive correlation indicated a strategy by which the number of pollen grains per plant could be enhanced by increasing all stages of the reproductive growth in the plant" in lines 122-123.
Lines 115-118: “The finding that the plants 'invested' more in pollen quality than in quantity in a year of reduced pollen production, such as 2017, may provide a clue to a strategy that allows the plants to compensate, to some extent, for the reduced level of pollen production.” This is an interesting topic which may apply to other biochemical processes as well. There is some indication that the amount of pollen allergens in “fixed at tree level”, i.e., the more pollen produced, the smaller the allergen content of pollen grains.
- Our assumption described above is based on works that reported the relationship between "production" and "quality" of pollen grains, e.g., the recent work of Pers-Kamczyc et al. [2020 Journal of Plant Physiology (https://doi.org/10.1016/j.jplph.2020.153156)]. They found that a higher availability of nutrients increases the production but decreases the quality of pollen grains in Juniperus communis L. when investigating the relationship between fertilization level and pollen productivity. We added the following sentence in lines 134-136 in the manuscript as well as the reference:
"Just as environmental and growth conditions affect pollen traits, so are nutrient levels linked to pollen quality and productivity, as has recently been reported in Juniperus communis by Pers-Kamczyc et al. [10]."
Table 3: please add information on the number of correlation tests (N).
- (n =11) was added in line 138.
I would have liked to read the authors’ thoughts about future research aspects. Especially discussion on (i) how male*female gametophytic interaction /compatibilities (/female choice) might make the breeding programmes more complex; and (ii) how to assure a wide enough genetic variation for future breeding needs.
- Now that we have shown the potential of our methodology for identifying prospective elite male cultivars, we could expand our research over a much broader range of males, under several climatic regions. This would provide a much more reliable depiction of the reproductive potential of jojoba males. Concurrently, as we identify prospective males with elevated EPP and desirable bloom phenologies, we intend to embark on a breeding program that would test the males against at least three of our best performing female cultivars. We intend to focus not only on pollen availability but also on pollen germinability, fruit and seed set percentages, and wax chemical traits (i.e. xenia and metaxenia effects). In a previous study on a very small scale (Vaknin et al., 2003), Dr. Yiftach Vaknin, a co-author of this manuscript, has shown that pollen source could significantly affect both reproductive success (seed set) and wax chemical constitution. We added this in lines 155-163.

Reviewer 2 Report
In this Manuscript, Zur et al. characterized the reproductive performance (flowering phenology, pollen production and pollen viability) of a group of male genotypes of jojoba. This study provides some insights into the strategy for identifying male genotypes with high-quality pollen production through the extended female flowering season.
The paper is well written, but there are some aspects of the manuscript that require further clarifications. Also, I have some concerns and questions:
- Lines 32-33: It is not clear to me. In my opinion, this paragraph needs some clarifications. Jojoba is a dioecious species. So, how do authors know which seed will give rise to a female or male plant? You cannot know the sex of the plant until it is fully developed unless you have a molecular marker linked to the sex trait. And you use it to identify the sex of the plant.
- Line 40: The format for reporting in which years the study was conducted is confusing. 2016-8?? This is not clear to me.
- I have some questions regarding the male genotypes used in this study. In lines 40-41, authors mentioned that the study was conducted on 8 and 25-year-old male genotypes (n = 150 and 150, respectively). However, in the result section authors presented data for 21 genotypes only. So, how many genotypes did authors use in the study? Why did authors select these 21 genotypes for conducting the study? How old were the 21 genotypes used for characterizing the flowering and floral traits? Did authors find any difference in terms of reproductive performances (flowering phenology, pollen production and pollen viability) between the 8 and 25 year-old male genotypes?
- Lines 45-47: How did authors set the peak flowering of the female plants? It would be nice if authors explain it in the manuscript.
- Line 56: n=6-12. What is this sample size for? Number of anthers per flower? It is not clear in the manuscript.
- Lines 59-60: How was calculated the EPP (effective pollen productivity)? I think authors should provide more details.
- Lines 60-62: On what basis did authors selected the 11 best performing genotypes? They reported it was based on flowering phenology and pollen production data of 2016. In the result section authors do not provide any further information about it. I guess they made their decision based on different traits. But it would be helpful if they provide some more details.
- Line 79: What do authors mean by good reproductive traits? On what basis did authors decide what genotypes had good reproductive traits? Did authors use any threshold for each of the traits analyzed?
- Line 83: Authors mentioned six genotypes with constant and reliable flowering period. However, in Table 1 they only highlighted five lines that were selected for future vegetative propagation. Why did authors discard genotype 40-51? Based on data from Table 1, it seems to me that the performance of 40-51 is similar or even better than the performance of 40-14 and 35-03.
- Table 1: Did authors perform analyses to test significant differences among genotypes within year? It would be helpful to have that data in Table 1.
- According to authors experience, what is the best or most accurate trait/traits for selecting males genotypes? Do authors think that measuring only the pollen viability could be enough for selecting males with high-quality pollen? What is their recommendation for other researcher interested in selecting elite male genotypes? What do authors think is the best strategy? It would be nice if authors could address those points in the manuscript.
Author Response
Reviewer #2
In this Manuscript, Zur et al. characterized the reproductive performance (flowering phenology, pollen production and pollen viability) of a group of male genotypes of jojoba. This study provides some insights into the strategy for identifying male genotypes with high-quality pollen production through the extended female flowering season.
The paper is well written, but there are some aspects of the manuscript that require further clarifications. Also, I have some concerns and questions:
Lines 32-33: It is not clear to me. In my opinion, this paragraph needs some clarifications. Jojoba is a dioecious species. So, how do authors know which seed will give rise to a female or male plant? You cannot know the sex of the plant until it is fully developed unless you have a molecular marker linked to the sex trait. And you use it to identify the sex of the plant.
- Female jojoba plants are clonally propagated (by rooting cuttings). Male plants are grown from seeds, i.e., seeds are germinated, plants are grown in pots until the first flowering season and only then are male plants transferred to the orchard. This is the current routine used by farmers. No molecular markers are used to identify the sex of the plant. We added this information in lines 33-35.
Line 40: The format for reporting in which years the study was conducted is confusing. 2016-8?? This is not clear to me.
- This research was performed from 2016 to 2018, i.e. 3 years. We clarified this point in the manuscript, line 46.
I have some questions regarding the male genotypes used in this study. In lines 40-41, authors mentioned that the study was conducted on 8 and 25-year-old male genotypes (n = 150 and 150, respectively). However, in the result section authors presented data for 21 genotypes only. So, how many genotypes did authors use in the study? Why did authors select these 21 genotypes for conducting the study? How old were the 21 genotypes used for characterizing the flowering and floral traits? Did authors find any difference in terms of reproductive performances (flowering phenology, pollen production and pollen viability) between the 8 and 25 year-old male genotypes?
- We performed a visual screen/characterization of 150 8-year-old and 150 25-year-old male genotypes (a total of 300) and chose to focus our research on 21 of these genotypes (see Figure 1). Among the 21 genotypes, 18 were 8 years old and 3 were 25 years old at the beginning of the research, year 2016. We added this information to Table 1.
Eleven genotypes were selected for the second year of study, based of pollen performances and flowering times (4, 4 and 3 for early, middle and late flowering). All these 11 genotypes belong to the 8-year-old group.
Regarding the last question on reproductive performances between the 8 and 25 year-old male genotypes – this question; i.e., fertility in younger versus older jojoba male plants, is beyond the scope of the current report and was not tested accordingly, therefore, without further investigation targeting age related traits we are not able to provide an answer to this comment.
Lines 45-47: How did authors set the peak flowering of the female plants? It would be nice if authors explain it in the manuscript.
- The peak of the flowering season in female cultivars is strongly affected by climatic conditions and may change between years. Most female cultivars bloom between middle of February and late March, but some bloom early, and some late. In the past five years blooming has been delayed and prolonged for most cultivars, probably due to the late arrival of spring or global warming. This variability in female blooming and the potential of poor overlap with male blooming motivated this research and draws attention the importance of identifying males with early, middle and late blooming characteristics. We added this paragraph in lines 35-40.
Line 56: n=6-12. What is this sample size for? Number of anthers per flower? It is not clear in the manuscript.
- Each flower was a biological replicate (6 to 12 flowers per genotype). At anthesis all the anthers/flower were transferred to a micro-tube. We added this in lines 61-63: 6 to 12 flowers per studied male genotype were harvested before stamen dehiscence, and at anthesis the anthers of each flower were transferred to a micro-tube containing 300 µl of 0.5 M sucrose (n = 6–12).
Lines 59-60: How was calculated the EPP (effective pollen productivity)? I think authors should provide more details.
- EPP was calculated as number of viable pollen grains per 1-m branch. It was calculated as the number of inflorescences per 1-m branch × the number of flowers per inflorescence × the number of pollen grains per flower × the percentage of viable pollen grains. We added more information in lines 68-69.
Lines 60-62: On what basis did authors selected the 11 best performing genotypes? They reported it was based on flowering phenology and pollen production data of 2016. In the result section authors do not provide any further information about it. I guess they made their decision based on different traits. But it would be helpful if they provide some more details.
- We selected male genotypes according to fertility parameters and flowering season: 4, 4 and 3 for early, middle and late flowering season. Consistency in the flowering season was most vital.
Line 79: What do authors mean by good reproductive traits? On what basis did authors decide what genotypes had good reproductive traits? Did authors use any threshold for each of the traits analyzed?
- We agree that the word "good" is not a clear scientific term, therefore, we replaced it with "above average," suggesting that our selected genotypes performed better and were better pollen producers than the general population of males in the orchard and some of the males that we tested.
Line 83: Authors mentioned six genotypes with constant and reliable flowering period. However, in Table 1 they only highlighted five lines that were selected for future vegetative propagation. Why did authors discard genotype 40-51? Based on data from Table 1, it seems to me that the performance of 40-51 is similar or even better than the performance of 40-14 and 35-03.
- Indeed six genotypes showed constant and reliable flowering seasons. Genotype 40-51 – one of these six genotypes - was not selected because of its lower number of flowers/inflorescence in comparison to 40-14 and 35-03.
Table 1: Did authors perform analyses to test significant differences among genotypes within year? It would be helpful to have that data in Table 1.
- As suggested, we performed analyses to test significant differences among genotypes in 2016.
For Materials and Methods:
"We used one-way analysis of variance (ANOVA) to compare EPP values of all tested genotypes in 2016 and in 2017. Post-hoc Tukey-Kramer HSD test was performed on ANOVA to compare the means in case of significant effect."
For Results and Discussion:
Values of EPP both in 2016 and in 2017 varied significantly among the tested genotypes (P<0.0001). See the tables below.
According to our results there is a huge difference between the 21 studied genotypes in 2016, indicating the need for this preliminary step before starting a comprehensive selection program. We will include this information and the tables upon editor's or reviewer's request. See below the calculations.
2016
2017
According to authors experience, what is the best or most accurate trait/traits for selecting males genotypes? Do authors think that measuring only the pollen viability could be enough for selecting males with high-quality pollen? What is their recommendation for other researcher interested in selecting elite male genotypes? What do authors think is the best strategy? It would be nice if authors could address those points in the manuscript.
- Our study revealed that the best strategy to identify males as potential elite cultivars is not to focus on a single reproductive trait, such as pollen viability or number of inflorescences, but to acquire data on all aspects of their reproductive traits as a means to providing a more accurate measure of their reproductive potential. We found that availability of viable pollen in a particular year was directly dependent on all the reproductive traits in that year, and on most – but not all – in the subsequent year (Table 3). A study of multiple reproductive traits – versus only a single trait, such as pollen viability – will enable the breeder to consistently and reliably identify elite males as future pollen providers. See lines 147-163.
